# Dyn-O: Building Structured World Models with Object-Centric Representations

**Zizhao Wang**[1,2*]    **Kaixin Wang**[2]    **Li Zhao**[2]    **Peter Stone**[1,3]    **Jiang Bian**[2]
[1]The University of Texas at Austin    [2]Microsoft Research Asia    [3]Sony AI
zizhao.wang@utexas.edu,pstone@cs.utexas.edu
{kaixwang,lizo,jiang.bian}@microsoft.com

## Abstract

World models aim to capture the dynamics of the environment, enabling agents to predict and plan for future states. In most scenarios of interest, the dynamics are highly centered on interactions among objects within the environment. This motivates the development of world models that operate on object-centric rather than monolithic representations, with the goal of more effectively capturing environment dynamics and enhancing compositional generalization. However, the development of object-centric world models has largely been explored in environments with limited visual complexity (such as basic geometries). It remains underexplored whether such models can be effective in more challenging settings. In this paper, we fill this gap by introducing Dyn-O, an enhanced structured world model built upon object-centric representations. Compared to prior work in object-centric representations, Dyn-O improves in both learning representations and modeling dynamics. On the challenging Procgen games, we demonstrate that our method can learn object-centric world models directly from pixel observations, outperforming DreamerV3 in rollout prediction accuracy. Furthermore, by decoupling object-centric features into dynamic-agnostic and dynamic-aware components, we enable finer-grained manipulation of these features and generate more diverse imagined trajectories. The code of Dyn-O can be found at: https://github.com/wangzizhao/dyn-O.

## 1   Introduction

World models have emerged as powerful tools for simulating environment dynamics, enabling agents to predict and plan for the future [16, 17, 20, 21, 39]. A common design in these models is to map high-dimensional observations (*e.g.*, images) into latent features and then model the environment's dynamics in this latent space. However, these latent features are typically monolithic, encoding the entire scene as a whole without accounting for its internal compositional structure. Yet, interactions in most environments are inherently object-centric. This observation motivates the development of object-centric world models, which can offer improved efficiency, interpretability, and compositional generalization.

Building object-centric world models involves two key steps: 1) learning object-centric representations and 2) modeling dynamics on top of them, as illustrated in Figure 1. For the former, the goal is to encode features associated with each object present in the observation. For the latter, the dynamics model should account for the different representation space, in contrast to a monolithic representation, the input and the prediction target is a set of object-centric representations. While several prior approaches have explored object-centric world models, they primarily focus on simple environments consisting of basic shapes [2, 13, 32, 42], or rely on externally provided compositional

---

[*]work done during an internship at Microsoft Research Asia

39th Conference on Neural Information Processing Systems (NeurIPS 2025).

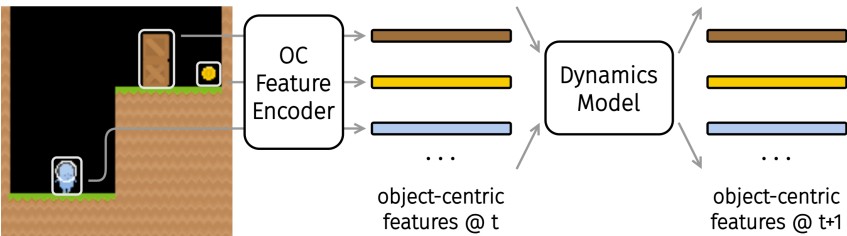

Figure 1: A high-level overview of the object-centric (OC) world model framework. The latent features are not monolithic or patch-based, but instead are bound to the objects present in the scene.

signals such as language [46]. It remains underexplored whether object-centric world models can be learned purely from trajectories to capture complex dynamics in more challenging, complex settings.

As a step toward bridging this gap, we introduce **Dyn-O**, a novel object-centric world model with improved designs for both object-centric representation learning and dynamics modeling over prior work. To handle complex visual observations, we draw inspiration from prior work on learning object-centric representations from real-world videos (*e.g.*, SOLV [1]) and adopt an autoencoder-style architecture to embed observations into object-associated features, referred to as *slots*. Specifically, we adopt a pre-trained Cosmos encoder [12], which offers two key advantages compared to the DINO encoder [34] used in SOLV: (1) improved representation quality, and (2) access to a pretrained visual decoder, eliminating the need to train a decoder from scratch for reconstructing pixel observations. To further enhance the quality of the extracted slot features, we incorporate priors from a high-performing pretrained segmentation model, SAM2 [35]. While this incorporation significantly improves performance, it also introduces substantial computational overhead, particularly during inference. To mitigate this issue, we use a scheduling strategy that gradually reduces reliance on the segmentation mask during training, enabling the model to maintain performance at inference time without requiring the mask.

The extracted slot features are modular in nature, with each slot associated with a distinct object. To model transition dynamics in this slot space, we aim for the world model to respect this modular structure. To this end, we adopt a state-space model (SSM) based on the Mamba architecture [15], borrowing the idea from SlotSSM [23]. Unlike SlotSSM, which tightly integrates the slot encoder and decoder within each transformer block of the SSM, we use the pretrained slot representation module introduced above and apply the SSM solely for dynamics modeling. This decoupling of representation learning and dynamics modeling aligns with recent trends in the world model literature [9, 29, 30]. Additionally, we explore disentangling each object's slot feature into a static component capturing time-invariant properties (*e.g.*, texture) and a dynamic component encoding time-varying properties (*e.g.*, position). This disentanglement enables fine-grained manipulation of slot features. For instance, we can modify the static feature of one object while keeping its dynamics unchanged, allowing for diverse data generation during world model rollouts.

We evaluate Dyn-O in seven Procgen [7] environments. Our experiments indicate that Dyn-O learns high-quality object-centric representations, generalizable world models, and disentangled static and dynamic representations. We summarize our contributions as follows.

- We propose a novel object-centric representation learning method by leveraging segmentation masks with a dropout schedule, enhancing representation quality while keeping efficient inference.
- We propose a novel object-centric world model that uses state-space models as backbones and outperforms monolithic models in both rollout quality and generalization.
- We propose a novel object-centric representation that disentangles each object's representation into static and dynamic features, allowing the generation of diverse rollouts by altering static attributes while preserving dynamic behavior.

## 2 Related Work

**World Models**    Accurately and efficiently modeling the world's dynamics has been a long-standing topic in reinforcement learning [39]. A learned model of a simulated environment can be used to

facilitate planning or generate imagined data for training a policy. In recent years, world models have seen great progress powered by advances in deep neural networks. Ha and Schmidhuber introduce a generative world model where the dynamics is learned entirely in a latent feature space. In a similar fashion, the Dreamer line of work [17–19] has further advanced the performance of world models. More recently, several works have explored using sequence modeling techniques to train world models within a token representation space [9, 29, 30].

Our works differs in that we build a world model on top of an object-centric representation, while prior work mainly focuses on monolithic representations. Closely related to our work are those that also consider the compositional nature of the world. Specifically, HOWM [45] and Cosmos [36] design object-centric world models to address the compositional generalization problem. DreamWeaver [2] uses a novel Recurrent Block-Slot Unit to discover compositional representations and generate compositional future simulations. However, those approaches mainly focus on small-scale diagnostic environments such as 2D block pushing, without validating in more complicated environments. RoboDreamer [46] learns a compositional world for robot manipulation tasks, but their method relies on language and leverages its natural compositionality. In comparison, our method learns the object-centric world model purely from agent trajectories.

**Object-Centric Representations**  Object-centric representations have gained increasing attention in recent years [6, 11, 22, 28, 37]. Instead of encoding an image observation as a single monolithic latent vector, object-centric representation learning aims to decompose a scene into objects and to learn different latent representations for each object. A key milestone in this direction was Slot Attention [28], which has since become the foundation for many subsequent methods that learn object-centric features in an unsupervised manner from images [22, 37, 43] or videos [1, 10, 25].

Our work builds on the SOLV framework [1], introducing improvements to enhance the extraction of object-specific latent features. Another closely related approach is the Slot State Space Model (SlotSSM) [23], which uses state-space models to capture long-range temporal dependencies in a structured way. However, it has only been validated on domains with limited complexity, such as basic shapes. Moreover, SlotSSM tightly couples slot representation learning and dynamics modeling, training them jointly. In contrast, we first learn a strong slot feature encoder and then build object-centric world models on top of it.

**Disentangling Static and Dynamic Features**  The disentanglement between static and dynamic features in sequences has been studied in computer vision [5, 31, 38]. DSVAE [44] divides the latent representation into the static factors and the dynamic factors and learns them jointly with the ELBO objective. However, as formalized by Locatello et al. [27], unsupervised disentanglement is impossible to achieve without inductive biases. C-DSVAE [3] further adds contrastive learning to encourage disentanglement, yet its learned dynamic features highly depend on the chosen data augmentations and can still capture static information. ContextWM [41] incorporates the notion of time-invariant context and time-varying latent variables into world models. However, using the same ELBO objectives as DSVAE, it may still learn entangled representations.

## 3  Method

From a high level perspective, Dyn-O models the environment's dynamics in an object-centric manner, enabled by two learning phases (Figure 2).

- **Object-centric representation learning** (Section 3.1): Dyn-O learns an object-centric representation in which coherent objects are each encoded into independent features, referred to as *slots*. This factorized representation, in contrast to a monolithic scene-level encoding, leverages the natural compositionality of objects in the world.
- **Dynamics learning** (Section 3.2): Dyn-O adopts a State Space Model (SSM) to predict transitions from the current slots to the next-step slots, conditioned on the action. Each object's slot feature is further decoupled into static and dynamic components to enable fine-grained manipulation.

### 3.1  Extracting Object-Centric Representations

In the first learning phase, Dyn-O learns an object-centric representation from image observations. In contrast to a monolithic representation that mixes all objects' information, this factored representa-

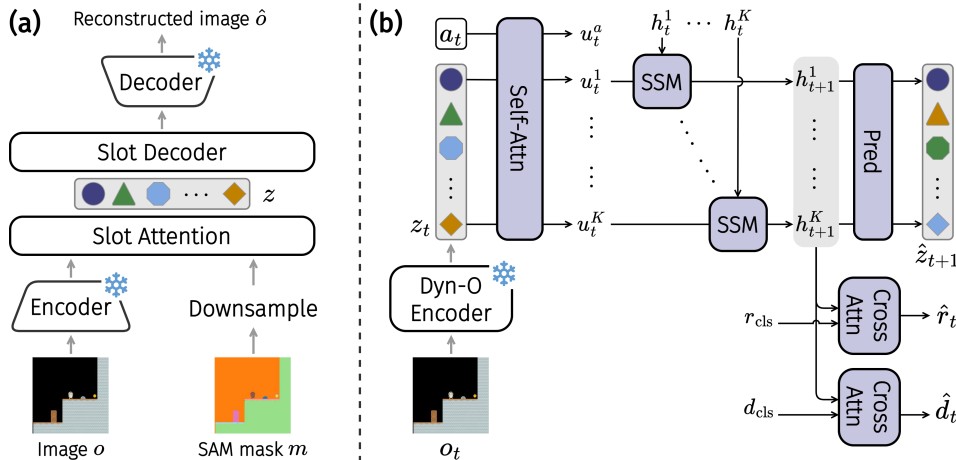

Figure 2: Components of Dyn-O: **(a)** object-centric representation learning; **(b)** dynamics learning. Modules marked with ❄ are fixed, while others are learnable. The "Dyn-O Encoder" in (b) corresponds to the lower half of (a), which maps the image $o$ to the latent slot feature $z$. See Section 3 for details.

tion enables Dyn-O to modify each object independently and generate novel object combinations. Formally, as shown in Figure 2(a), Dyn-O learns an encoder $\texttt{Enc} : \mathcal{O} \to \mathcal{Z}$ to extract an object-centric representation $z \in \mathcal{Z}$ from observations $o$, where the representation consists of $K$ slots, $z = [z^1, \ldots, z^K]$, each corresponding to an object. Here, $K$ is a predefined hyperparameter, and when the number of objects in a scene is smaller than $K$, some slots may remain unused.

During encoder learning, inspired by SOLV [1], instead of training from scratch with raw pixels, Dyn-O learns on top of the high-quality features extracted by the Cosmos tokenizer [12]. Given an input frame $o \in \mathbb{R}^{H \times W \times 3}$, Dyn-O first applies the Cosmos encoder ($\texttt{CosmosEnc}$) to extract patch-level features $f \in \mathbb{R}^{N \times d_f}$, where $N$ denotes the number of patches in the frame. It then initializes $K$ slots $z_{\text{init}} \in \mathbb{R}^{K \times d_z}$ from a set of learnable vectors. These slots compete to bind to objects using iterative Slot Attention [28], producing $z = \texttt{Slot-Attn}(z_{\text{init}}, f)$. The learning signals for slot extraction arise from reconstructions – a decoder ($\texttt{Slot-Dec}$) reconstructs features $\hat{f}$ from the slots, which are then used to reconstruct the observation via the Cosmos decoder as $\hat{o} = \texttt{CosmosDec}(\hat{f})$. That is,

$$f = \texttt{CosmosEnc}(o),$$
$$z = \texttt{Slot-Attn}(z_{\text{init}}, f),$$
$$\hat{f} = \texttt{Slot-Dec}(z),$$
$$\hat{o} = \texttt{CosmosDec}(\hat{f}).$$

The $\texttt{Slot-Attn}$ and $\texttt{Slot-Dec}$ modules are trained jointly by minimizing the reconstruction error as follows, while the Cosmos encoder-decoder remains frozen:

$$\mathcal{L}_{\text{slot}} = \|f - \hat{f}\|^2 + \|o - \hat{o}\|^2. \tag{1}$$

Empirically, we observe that this fully unsupervised training objective usually results in inaccurate object-slot bindings (see examples in Section 4.1). To improve the quality of object-centric representations, we leverage the prior provided by foundational segmentation models. Specifically, we use a pre-trained SAM2 model [35] to generate a segmentation mask $m \in \{0, 1\}^{H \times W \times K}$ and use it as an attention mask during slot attention, $z = \texttt{Slot-Attn}(z_{init}, f, m)$, binding each slot to one segmented object and constraining it to only attend to patches from that object.

While the segmentation mask enhances object-slot binding, it also introduces substantial computational overhead, especially during inference. This dependency could limit the practicality of Dyn-O. For example, when extracting slots for an agent interacting with the environment in an online setting, we would need to run SAM2 inference at every environment step, which would be prohibitively expensive compared to the typical cost per step. To address this issue, during encoder learning, we

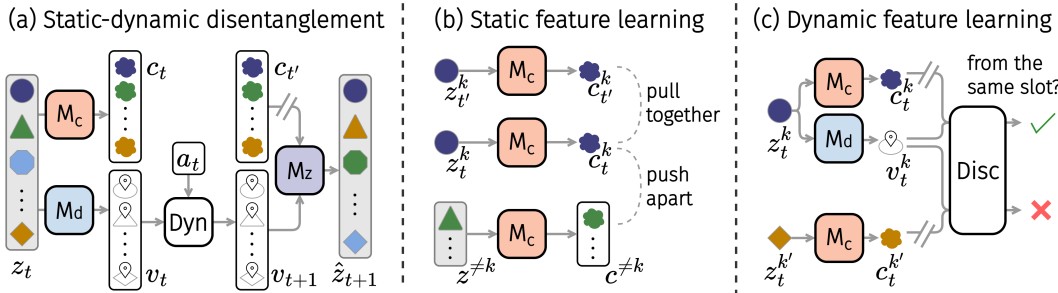

Figure 3: Illustration of **(a)** the overall design for disentangling slot features in dynamics modeling, and the training procedures for **(b)** static features and **(c)** dynamic features. // indicates a stop-gradient operation.

introduce an annealing schedule that gradually reduces the reliance on the segmentation mask, aiming to eliminate its use by the end of training. Initially, the segmentation mask is always used to guide slot extraction. As training progresses, the probability of not using the mask increases according to a logarithmic schedule w.r.t. the number of network updates, as $\log(1+\#\text{ updates})/\log(1+\#\text{ total updates})$. This dropout schedule allows the encoder to achieve the best of both worlds – it learns high-quality representations with the initial guidance of segmentation masks, while enabling efficient inference by phasing out the need for them.

## 3.2 World Model with Object-Centric Representations

After learning object-centric representations as described above, the next phase of Dyn-O focuses on training a world model to reason about object interactions. Given the history of slots $z_{\leq t}$ and actions $a_{\leq t}$, the world model predicts the next slots, the reward, and whether the episode terminates as $\hat{z}_{t+1}, \hat{r}_t, \hat{d}_t = \text{Dyn}(z_{\leq t}, a_{\leq t})$.

**World Model**  For the world model Dyn, we adopt state-space models (SSMs) for their strength at capturing long-range temporal dependencies. Meanwhile, the model needs to account for the *permutation equivariance* among slots – if the input slots are randomly permuted, the slots' prediction should follow the same permutation. Therefore, as shown in Figure 2(b), we design Dyn as follows. We first apply self-attention to extract information about object interactions:

$$u_t^k = \text{Self-Attn}(\text{q} = z_t^k, \text{kv} = [z_t^1, \ldots, z_t^K, a_t]),$$

where no position encoding is used to maintain permutation invariance. Next, each slot is processed by a shared SSM to update its hidden state, which is then used by Dyn-O to predict the next slots, reward, and whether the episode terminates using separate prediction networks as follow:

$$h_{t+1}^k = \text{SSM}(h_t^k, z_k^t), \quad \hat{z}_{t+1}^k = \text{Pred}(h_{t+1}^k),$$
$$\hat{r}_t = \text{Cross-Attn}(\text{q} = r_{\text{cls}}, \text{kv} = [h_{t+1}^1, \ldots, h_{t+1}^K]),$$
$$\hat{d}_t = \text{Cross-Attn}(\text{q} = d_{\text{cls}}, \text{kv} = [h_{t+1}^1, \ldots, h_{t+1}^K]),$$

where $h^k$ is the hidden state of the SSM that tracks past information for the $k$-th slot, and $r_{\text{cls}}$ and $d_{\text{cls}}$ learnable query tokens used to extract reward and termination signals from the hidden states. Finally, the world model is optimized by minimizing the following prediction loss (Alg. 1 line 4) :

$$\mathcal{L}_{\text{wm}} = \sum_{t=1}^{T-1} \|\hat{z}_{t+1} - z_{t+1}\|^2 + \sum_{t=1}^{T} \left( \|\hat{r}_t - r_t\|^2 + \text{CE}(\hat{d}_t, d_t) \right), \tag{2}$$

where CE stands for cross-entropy loss for binary classification.

**Static-Dynamic Disentanglement**  The object-centric representation learned in Section 3.1 enables the creation of scenes with novel object combinations. However, when synthesizing data, it is often desirable to only modify object appearance (*e.g.*, colors) while preserving their dynamic properties

(*e.g.*, positions). For example, modifying dynamic information could result in invalid scenes, such as a table being moved while the objects originally on it remain suspended in midair.

To this end, as shown in Figure 3(a), Dyn-O disentangles each slot $z^k$ into two components: static features $c^k \in \mathbb{R}^{d_c}$ that captures time-invariant properties and dynamic features $v^k \in \mathbb{R}^{d_v}$ that encodes time-variant properties. This decomposition allows us to modify an object's static features while keeping its dynamic features unchanged, generating novel scenarios with consistent dynamics. Formally, Dyn-O maps slots to static and dynamic features with two separate neural networks: $c^k = \mathtt{M}_c(z^k)$ and $v^k = \mathtt{M}_v(z^k)$. We describe how these features are learned below.

Static features are intended to capture time-invariant properties. Thus, for an object present at timestep $t$, its static feature should remain the same across all timesteps. Therefore, a natural training objective is to minimize the difference between static features across all timesep pairs. However, this objective alone admits a degenerate solution where all static features collapse to a constant. To prevent this collapse, Dyn-O incorporates contrastive learning to encourage feature diversity, as illustrated in Figure 3(b): static features should be similar if they belong to the same slot, and distinct otherwise. Dyn-O therefore learns static features by optimizing the following loss (Alg. 1 line 5):

$$\mathcal{L}_{\text{stat}} = \underbrace{\sum_{k, t \neq t'} \|c_t^k - c_{t'}^k\|^2}_{\text{time invariance}} + \underbrace{\sum_{k,t} -\log \frac{\cos(c_t^k, c_{t'}^k)}{\cos(c_t^k, c_{t'}^k) + \sum_{\tilde{c} \in \tilde{C}} \cos(c_t^k, \tilde{c})}}_{\text{contrastive learning}}, \tag{3}$$

where $\cos$ denotes cosine similarity, and the second term corresponds to the InfoNCE loss [33]. For each $c_t^k$, the positive sample $c_{t'}^k$ comes from the same slot at a different timestep, while the negative samples $\tilde{C}$ are drawn from other slots in the batch.

On the other hand, Dyn-O learns dynamic features by reconstructing the slot content while ensuring their disentanglement from static features. Specifically, Dyn-O uses a reconstruction network $\mathtt{M}_z$ to output $\hat{z}^k = \mathtt{M}_z(\mathtt{sg}(c^k), v^k)$, where $\mathtt{sg}$ denotes stop-gradient, preventing the static feature $c^k$ from being updated by the reconstruction loss. (For simplicity, in this paragraph, we omit the subscript $t$ in $z, c$ and $v$). However, minimizing reconstruction loss alone may lead the dynamic features to encode time-invariant information as well, bypassing the need for $c^k$. To avoid this effect, Dyn-O promotes disentanglement by minimizing the mutual information $\sum_{t,k} I(c_t^k, v_t^k)$ via adversarial training, as illustrated in Figure 3(c). A discriminator $\mathtt{Disc}$ is trained to distinguish whether a pair of static and dynamic features comes from the same slot. While $\mathtt{Disc}$ minimizes the discrimination loss, the dynamic feature extractor $\mathtt{M}_v$ is trained to maximize it, thereby reducing the static information encoded in dynamic features [14]. For stability, we adopt the Wasserstein distance as the discrimination loss and apply LeCam regularization [40] (Alg. 1, lines 6-7):

$$\mathcal{L}_{\text{dyn}} = \sum_k \underbrace{\|\hat{z}^k - z^k\|^2}_{\text{reconstruction}} + \underbrace{\mathtt{Disc}(c^k, v^k)}_{\text{disentanglement}}, \tag{4}$$

$$\mathcal{L}_{\text{disc}} = \sum_k \underbrace{\left(-\mathtt{Disc}(c^k, v^k) + \mathtt{Disc}(c^k, d^{k'})\right)}_{\text{discrimination}} + \underbrace{\mathtt{LeCam}(\mathtt{Disc})}_{\text{regularization}}, \tag{5}$$

where $(c^k, v^k)$ are from the same slot, and $(c^k, d^{k'})$ are from different slots.

## 4 Experimental Evaluation

We evaluate Dyn-O to answer the following questions: Q1: Can Dyn-O learn accurate object-centric representations (Sec. 4.1)? Yes. Q2: Can Dyn-O learn accurate world models (Sec. 4.3)? Yes. Q3: Are static and dynamic features learned by Dyn-O truly disentangled (Sec. 4.4)? Yes.

Our experiments are conducted in **Procgen** [7], a set of procedurally-generated 2D video game environments. We use 7 Procgen environments: bigfish, coinrun, caveflyer, dodgeball, jumper, ninja, and starpilot. In each Procgen environment, a PPG policy [8] is trained and used to collect an offline dataset of 1M transitions from the first 200 levels for the learning of all methods.

**Algorithm 1** Dyn-O World Model Learning

---

1: Collect a dataset of $(o_t, a_t, r_t, o_{t+1})$. Initialize object-centric encoder (`Enc`), mappings to static features (`M`$_c$), dynamic features (`M`$_v$), and slots (`M`$_z$), the world model (`Dyn`), and the discriminator (`Disc`).
2: Train object-centric encoder `Enc` with Eq. (1).
3: Train the world model with joint optimization:
4:     Update the world model `Dyn` by minimizing prediction losses in Eq. (2).
5:     **if** learn static-dynamic disentanglement **then**
6:         Update static features from `M`$_c$ by enforcing time-invariance in Eq. (3).
7:         Update `Disc` with Eq. (4) to estimate mutual information.
8:         Update dynamic feature from `M`$_v$ via reconstruction `M`$_z$ and disentanglement loss in Eq. (5).

---

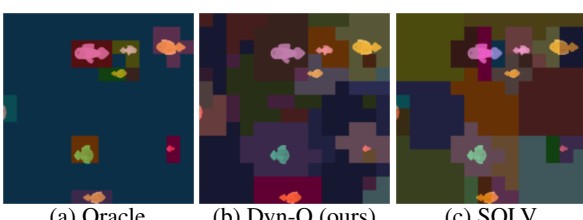

| Method | FR-ARI ($\uparrow$) |
| --- | --- |
| Oracle | 0.96 |
| Dyn-O (ours) | 0.80 |
| SOLV | 0.54 |

(a) Oracle     (b) Dyn-O (ours)     (c) SOLV     (d) slot-object binding accuracy

Figure 4: Evaluation of the object-centric representation learning in **bigfish**. **(a)** Using segmentation masks during inference, Oracle achieves the most accurate slot-object binding, but also suffering from the high computational overhead. **(b)** By using segmentation masks only during training, Dyn-O learns to accurately assign each fish to one slot (shown as colored patches), which avoiding inference-time overhead. **(c)** In contrast, learning without the guidance of segmentation masks, SOLV often inaccurately splits a fish into multiple separate slots.

## 4.1 Evaluating Object-Centric Representation

As Dyn-O's world model is learned on top of the object-centric representations, the quality of learned representation is critical to its prediction accuracy. We compare Dyn-O's representation learning against the following ablations:

- **Oracle**: our method but always using the segmentation mask during both training and inference.

- **SOLV** [1]: the same as our method but does not use segmentation mask for training and inference.

As shown in Fig. 4, compared to SOLV that often splits an object into multiple slots, Dyn-O achieves more accurate object-slot binding, assigning each object to a single slot. We also evaluate the foreground adjusted rand index (FG-ARI) which is a widely used metric in the object-centric literature that measures the similarity of the discovered objects masks to ground-truth masks. Again, Dyn-O outperforms SOLV, suggesting the benefit of using segmentation masks to guide representation learning.

## 4.2 Evaluating World Model Accuracy

The promise of Dyn-O is to learn accurate and and generalizable world models based on object-centric representations. Therefore, the most critical evaluation of our work focuses on the world model quality, and we compare Dyn-O against the following baselines and ablations:

- **DreamerV3** [19]: one of the state-of-the-art model-based RL methods.

- **Dreamweaver** [2]: an object-centric world model designed to discover hierarchical and compositional representations.

- **Dyn-O without object-centric representations** (denoted as **Dyn-O w/o OC**): our method but without object-centric representations. The dynamics model is learned on top of 14x14 patch-level features extracted by Cosmos encoder, where each patch is treated as a "slot".

For a fair comparison, we use the same frozen Cosmos tokenizer for DreamerV3 and Dreamweaver as Dyn-O during world model learning.

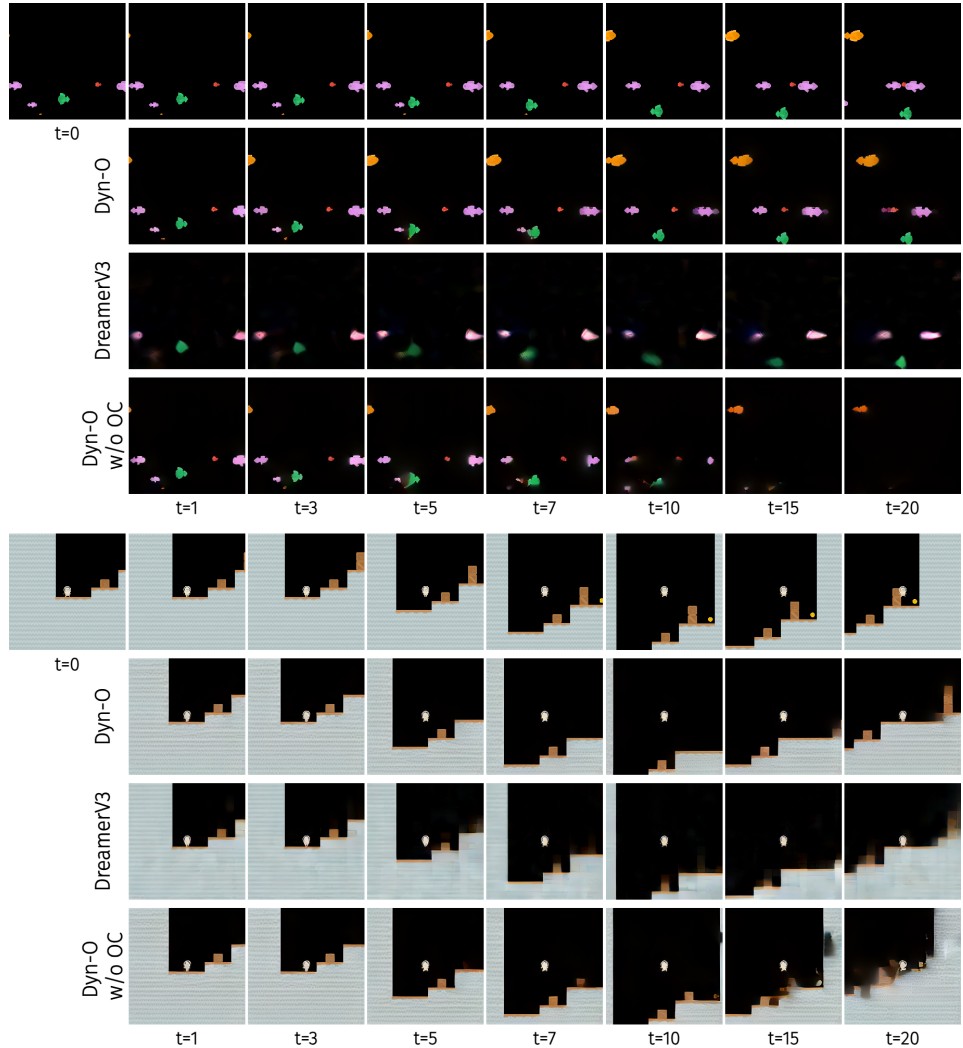

Figure 5: Dyn-O generates more accurate rollouts than Dreamer in **bigfish** and **coinrun**. In each environment, the first row displays a trajectory collected in the real environment. The second row depicts the prediction inside the world model by Dyn-O (ours). The third and fourth rows show the prediction of Dreamer and Dyn-O w/o OC respectively. In bigfish, our method keeps consistent prediction for each fish until the 15th step, while baselines lose track of multiple small fish before the 10th step. Similarly, in coinrun, compared to baselines, Dyn-O generates predictions with clearer floor and boxes.

To evaluate the generalizability of each method, we use the learned world model to generate 20-step rollouts in 500 unseen levels. The evaluation metric covers pixel-level quality (PSNR) as well as temporal and spatial coherence (FVD, LPIPS, and SSIM). The results are shown in Table 1, where we average the evaluation at the last step across all environments. The results for each environment can be found in Appendix B.3. Dyn-O significantly outperforms dreamer which uses a monolithic representation, demonstrating the advantage of predicting in the object-centric representation space. Meanwhile, in contrast to Dyn-O w/o OC whose latent representation consist of 196 patch-level "slots", Dyn-O only uses 31 or 47 object-level slots (where each object-level slot has the same dimension as a patch-level "slot") and achieves higher performance, demonstrating the superior efficiency of object-centric representations. The results for Dreamwaver can be found in Appendix B.3. To complement the analysis with qualitative examples, Fig. 5 shows the generated rollouts of Dyn-O and its comparison with baselines.

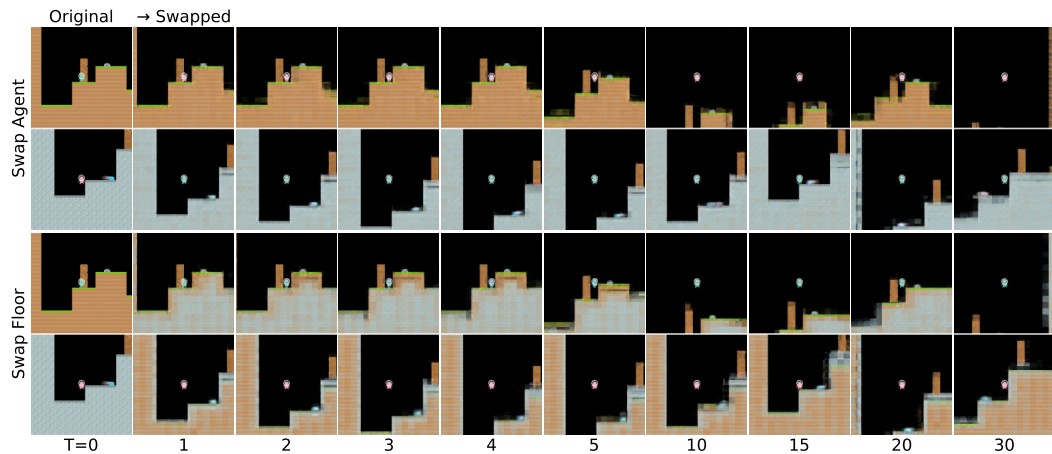

Figure 6: Dyn-O generates dynamically consistent rollouts after exchanging static features. In the top two rows, we swap the static features of the avatar (the small agent in the center of the image) between two initial states and generate 30-step rollouts using Dyn-O. The results show that only the avatar's color changes, while all other objects remain unchanged. In the bottom two rows, we exchange the static features of the floor, and Dyn-O consistently swaps their colors while keeping all other objects intact.

Table 1: Quantitative results for rollout trajectories, averaged across all environments.

| Method | LPIPS (↓) | FVD (↓) | SSIM (↑) | PSNR (↑) |
|---|---|---|---|---|
| DreamerV3 | 0.42 | 692.5 | 0.56 | 15.70 |
| Dyn-O w/o OC | 0.41 | 538.4 | 0.53 | 16.10 |
| Dyn-O (ours) | **0.33** | **361.3** | **0.62** | **16.34** |

In addition to Procgen, we further compare Dyn-O against DreamerV3 on the CLEVR dataset [24] and two ALE environments [4], and the results are in Appendix B.3.

## 4.3 Evaluating Representation Effectiveness for Policy Learning

Next, we evaluate whether the learned object-centric representations can facilitate policy learning. To do so, we fix the representation backbone and train only the policy head using the PPG algorithm on three Procgen games: Bigfish, Starpilot, and Coinrun, with three random seeds for each game. We use the "easy" difficulty setting with an unlimited number of levels.

We compare the performance of policies using object-centric representations learned by Dyn-O against policies using representations directly output by the Cosmos encoder, and the policy performance (measured as reward) are shown in Table. 3. Policies using object-centric representations achieve much higher reward than policies with raw patch-level features, demonstrating the effectiveness of Dyn-O's representations for downstream task learning.

Table 2: Probing accuracy (↑), in percentage (%), on **coinrun** privilege properties, shown the mean and standard deviation . Static features have much higher prediction accuracy than dynamic features for static properties (i.e., RGB values), while dynamic features have higher accuracy on dynamic properties (i.e., position and area), demonstrating that Dyn-O achieves effective static-dynamic disentanglement

| | Static Properties | | | Dynamic Properties | | |
|---|---|---|---|---|---|---|
| | R value | G value | B value | x position | y position | area |
| slots | **83.5** ± 0.0 | **81.7** ± 0.0 | **89.3** ± 0.0 | **91.8** ± 0.0 | **94.5** ± 0.0 | **97.1** ± 0.0 |
| dynamic features | 47.7 ± 7.3 | 45.7 ± 7.3 | 47.0 ± 8.9 | 78.1 ± 5.1 | 75.2 ± 6.6 | 83.3 ± 3.1 |
| static features | 67.3 ± 1.4 | 70.9 ± 1.9 | 81.7 ± 2.2 | 34.9 ± 1.2 | 37.7 ± 2.0 | 75.3 ± 0.4 |
| random features | 25.8 ± 0.0 | 27.3 ± 0.0 | 23.2 ± 0.0 | 29.3 ± 0.0 | 28.3 ± 0.0 | 73.3 ± 0.0 |

Table 3: Reward of policies using different representations, measured by the mean and standard deviation across three random seeds.

| Representation | Bigfish | Starpilot | Coinrun |
|---|---|---|---|
| Cosmos encoder | $0.90 \pm 0.02$ | $8.80 \pm 0.47$ | $8.00 \pm 0.18$ |
| Dyn-O (ours) | $\mathbf{7.67} \pm 4.02$ | $\mathbf{19.19} \pm 0.88$ | $\mathbf{8.95} \pm 0.38$ |

## 4.4 Evaluating Static-Dynamic Disentanglement

To examine whether the static and dynamic features learned by Dyn-O are disentangled, We probe the model with environment-privileged information, evaluating whether static features only capture time-invariant properties and whether dynamic features only capture time-varying properties. Specifically, we use SAM to segment out objects in 10K transitions and extract the following properties for each object as probing targets: area (i.e., the number of pixels), xy positions, and average RGB values across object pixels. Then we compute each object's **slot**, **static** feature, and **dynamic** feature as probing inputs. During probing, we discretize each property into 10 bins and use a linear classifier to predict the target. To establish a reference point for interpretation, we also perform the same prediction using randomly initialized features, representing the performance expected when no meaningful information is available in the inputs. Table 2 shows the prediction accuracy of all features in the coinrun environment (see results for other environments in the Appendix). Dyn-O shows strong disentanglement results, particularly as the prediction accuracy of static features for position and area is close to that of random features, indicating that static features capture little to no information about dynamic properties.

We further illustrate the disentanglement between static and dynamic features with qualitative examples. Fig. 6 presents rollouts generated after swapping static features between two initial states. As shown, rollouts from the same initial states (the first and third rows) remain nearly identical, except for the color changes in the avatar and floor. The same pattern is observed in the second and fourth rows, demonstrating that Dyn-O preserves dynamics while modifying only static properties.

## 5 Conclusion

We present Dyn-O, a world modeling method that builds on object-centric representations. By leveraging segmentation masks during training with a schedule, Dyn-O learns high-quality object-centric representations while avoiding the overhead of segmentation computation at inference time. Additionally, Dyn-O further disentangles each object feature into static, time-invariant components (e.g., color) and dynamic, time-variant components (e.g., position), allowing it to generate diverse predictions. Dyn-O predicts in latent space, with a pretrained decoder used for visualization. Exploring more sophisticated decoding methods, such as diffusion models, could further enhance the visual quality of generated rollouts.

## Acknowledgements

The majority of this work has taken place in the Learning Agents Research Group (LARG) at the Artificial Intelligence Laboratory, The University of Texas at Austin. LARG research is supported in part by the National Science Foundation (FAIN-2019844, NRT-2125858), the Office of Naval Research (N00014-24-1-2550), Army Research Office (W911NF-17-2-0181, W911NF-23-2-0004, W911NF-25-1-0065), DARPA (Cooperative Agreement HR00112520004 on Ad Hoc Teamwork), Lockheed Martin, and Good Systems, a research grand challenge at the University of Texas at Austin. The views and conclusions contained in this document are those of the authors alone. Peter Stone serves as the Chief Scientist of Sony AI and receives financial compensation for that role. The terms of this arrangement have been reviewed and approved by the University of Texas at Austin in accordance with its policy on objectivity in research.

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

Table 4: The Architecture and Hyperparameters of Encoder Learning.

| Name | Value | | | | | | |
|---|---|---|---|---|---|---|---|
| | bigfish | caveflyer | coinrun | dodgeball | jumper | ninja | starpilot |
| # slots | 31 | 63 | 31 | 31 | 31 | 47 | 47 |
| slot dimension | | | | 256 | | | |
| # slot attention iterations | | | | 3 | | | |
| image size | | | | [224, 224, 3] | | | |
| patch size | | | | 16 | | | |
| optimizer | | | | Adam | | | |
| learning rate | | | | 4e-4 | | | |
| data size | | | | 1M | | | |
| epoch | | | | 15 | | | |
| batch size | | | | 64 | | | |
| Cosmos model | | | | Cosmos-0.1-Tokenizer-CI16x16 | | | |

Table 5: The Architecture and Hyperparameters of Dynamics Learning.

| Name | Value |
|---|---|
| dynamic feature dimension | 256 |
| static feature dimension | 256 |
| self-attention # layers | 1 |
| self-attention model size | 512 |
| self-attention # heads | 8 |
| SSM # layers | 2 |
| SSM model size | 512 |
| SSM $d_{state}$ | 64 |
| SSM $d_{conv}$ | 4 |
| optimizer | Adam |
| learning rate | 1e-4 |
| batch size | 32 |

# A Method Details

## A.1 Encoder

The architecture and hyperparameter used during encoder training are shown in Table. 4.

## A.2 Dynamics

During dynamic feature training, the slot reconstruction loss and the disentanglement loss may have conflicting gradient and hinder model learning. To mitigate this issue, we leverage gradient modification [26] to enhance the balance between two objectives.

The architecture and hyperparameter used during encoder training are shown in Table. 5.

# B Experiment Details

## B.1 Environment Details

Among the 16 Procgen environments, we selected 7 environments based on their relevance to object-centric learning and visual complexity. Specifically, we looked for environments with **moving objects**, **dynamic layouts**, and **minimal sensitivity to color**. Based on these criteria, we excluded maze-like environments such as Maze, Heist, Chaser, and Miner, as well as Plunder, where the ship's color plays a critical role. Among the remaining games, we randomly selected 7 while ensuring diversity across game types, given resource constraints.

Table 6: slot-object binding accuracy, measured by FR-ARI (↑).

| Environments | Oracle | Dyn-O (ours) | SOLV |
|---|---|---|---|
| bigfish | 0.96 | 0.80 | 0.54 |
| coinrun | 0.33 | 0.27 | 0.10 |
| dogeball | 0.79 | 0.48 | 0.17 |
| starpilot | 0.86 | 0.47 | 0.49 |
| average | 0.74 | 0.51 | 0.33 |

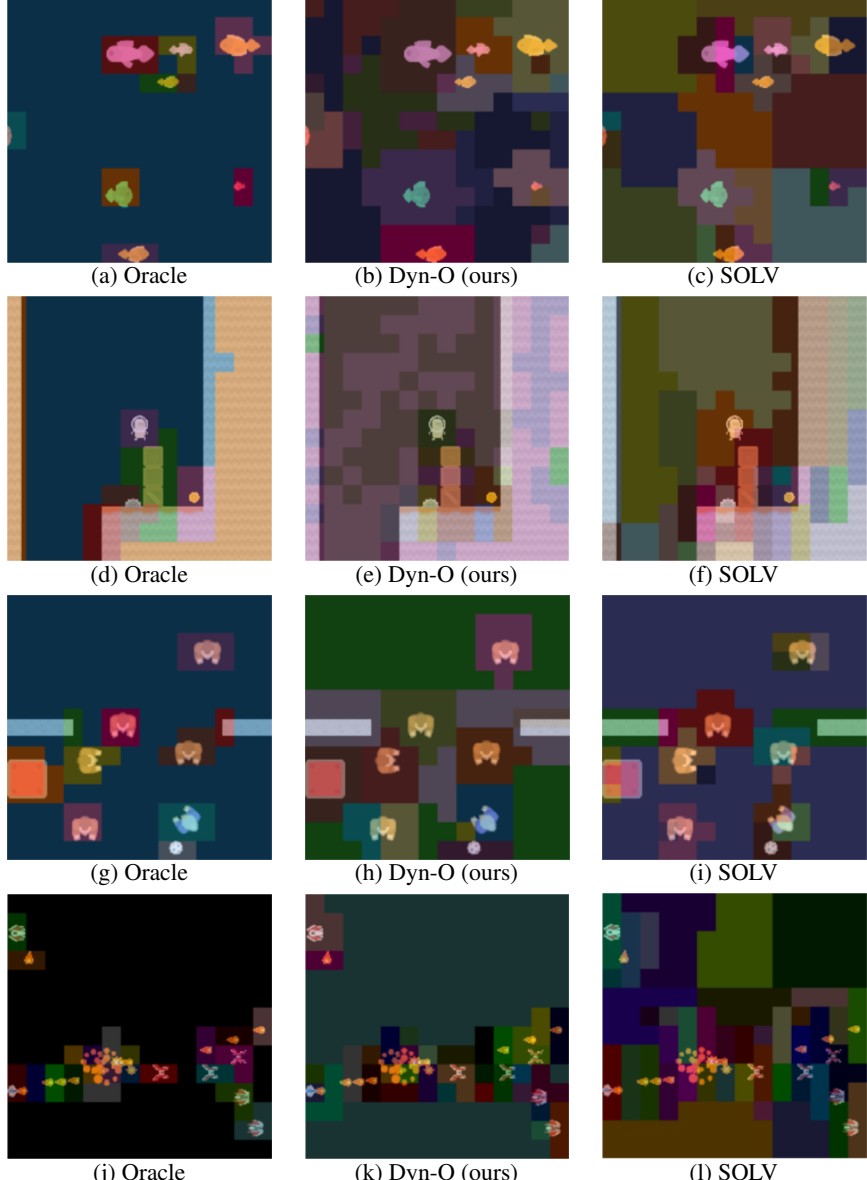

Figure 7: Qualitative evaluation of the object-centric representation learning in **bigfish**, **coinrun**, **dodgeball**, and **starpilot**.

## B.2 Evaluating Object-Centric Representation

The object-centric representation evaluation for 4 procgen environments are shown in Table. 6 and Fig. 7. By leveraging segmentation masks only during training, Dyn-O significantly outperforms SOLV in all environments, in terms of slot-object binding accuracy.

Table 7: Rollout accuracy for each Procgen environment at 20-th timestamp, measured as mean and standard error.

| Environment | Metric | DreamerV3 | Dreamerwaver | Dyn-O w/o OC | Dyn-O (ours) |
|---|---|---|---|---|---|
| bigfish | LPIPS (↓) | $0.39 \pm 0.01$ | $0.52 \pm 0.01$ | $0.36 \pm 0.01$ | $\mathbf{0.25} \pm 0.01$ |
| | FVD (↓) | $567.20 \pm 11.18$ | $831.17 \pm 15.55$ | $248.94 \pm 16.32$ | $\mathbf{126.88} \pm 6.42$ |
| | SSIM (↑) | $0.73 \pm 0.01$ | $0.68 \pm 0.00$ | $0.68 \pm 0.01$ | $\mathbf{0.76} \pm 0.01$ |
| | PSNR (↑) | $19.34 \pm 0.14$ | $19.09 \pm 0.12$ | $20.16 \pm 0.27$ | $\mathbf{20.28} \pm 0.30$ |
| caveflyer | LPIPS (↓) | $\mathbf{0.53} \pm 0.01$ | – | $0.59 \pm 0.01$ | $0.61 \pm 0.01$ |
| | FVD (↓) | $1104.50 \pm 18.87$ | – | $966.07 \pm 28.71$ | $\mathbf{877.74} \pm 20.99$ |
| | SSIM (↑) | $\mathbf{0.44} \pm 0.01$ | – | $0.26 \pm 0.01$ | $0.26 \pm 0.01$ |
| | PSNR (↑) | $\mathbf{11.99} \pm 0.12$ | – | $10.93 \pm 0.19$ | $10.82 \pm 0.13$ |
| coinrun | LPIPS (↓) | $0.34 \pm 0.01$ | – | $0.36 \pm 0.01$ | $\mathbf{0.28} \pm 0.01$ |
| | FVD (↓) | $530.31 \pm 10.51$ | – | $583.11 \pm 16.07$ | $\mathbf{266.13} \pm 6.16$ |
| | SSIM (↑) | $0.60 \pm 0.01$ | – | $0.47 \pm 0.01$ | $\mathbf{0.62} \pm 0.01$ |
| | PSNR (↑) | $\mathbf{14.22} \pm 0.25$ | – | $12.61 \pm 0.14$ | $13.56 \pm 0.19$ |
| dodgeball | LPIPS (↓) | $0.42 \pm 0.01$ | – | $0.15 \pm 0.01$ | $\mathbf{0.14} \pm 0.01$ |
| | FVD (↓) | $903.51 \pm 20.44$ | – | $481.55 \pm 20.83$ | $\mathbf{372.50} \pm 15.07$ |
| | SSIM (↑) | $0.50 \pm 0.01$ | – | $0.72 \pm 0.01$ | $\mathbf{0.76} \pm 0.01$ |
| | PSNR (↑) | $16.10 \pm 0.05$ | – | $20.33 \pm 0.12$ | $\mathbf{20.88} \pm 0.15$ |
| ninja | LPIPS (↓) | $0.48 \pm 0.01$ | – | $0.49 \pm 0.01$ | $\mathbf{0.45} \pm 0.01$ |
| | FVD (↓) | $423.27 \pm 14.26$ | – | $521.32 \pm 15.85$ | $\mathbf{313.57} \pm 9.73$ |
| | SSIM (↑) | $0.45 \pm 0.01$ | – | $0.33 \pm 0.01$ | $\mathbf{0.54} \pm 0.01$ |
| | PSNR (↑) | $12.58 \pm 0.17$ | – | $\mathbf{12.80} \pm 0.10$ | $11.95 \pm 0.12$ |
| starpilot | LPIPS (↓) | $0.37 \pm 0.01$ | $0.50 \pm 0.01$ | $0.50 \pm 0.01$ | $\mathbf{0.22} \pm 0.01$ |
| | FVD (↓) | $626.47 \pm 14.95$ | $735.24 \pm 21.48$ | $429.36 \pm 15.46$ | $\mathbf{211.27} \pm 12.70$ |
| | SSIM (↑) | $0.69 \pm 0.01$ | $0.68 \pm 0.00$ | $0.72 \pm 0.01$ | $\mathbf{0.76} \pm 0.01$ |
| | PSNR (↑) | $19.99 \pm 0.08$ | $19.48 \pm 0.13$ | $19.76 \pm 0.23$ | $\mathbf{20.55} \pm 0.13$ |

Table 8: Rollout accuracy for CLEVR and ALE environments at 20-th timestamp, measured as mean and standard error.

| Environment | Metric | DreamerV3 | Dyn-O (ours) |
|---|---|---|---|
| CLEVR | LPIPS (↓) | $0.34 \pm 0.00$ | $\mathbf{0.31} \pm 0.00$ |
| | FVD (↓) | $1676.18 \pm 40.92$ | $\mathbf{446.57} \pm 8.55$ |
| | SSIM (↑) | $0.86 \pm 0.00$ | $\mathbf{0.88} \pm 0.00$ |
| | PSNR (↑) | $21.37 \pm 0.09$ | $\mathbf{22.63} \pm 0.06$ |
| Atari Skiing | LPIPS (↓) | $0.12 \pm 0.00$ | $\mathbf{0.09} \pm 0.01$ |
| | FVD (↓) | $217.84 \pm 3.76$ | $\mathbf{187.76} \pm 8.04$ |
| | SSIM (↑) | $0.91 \pm 0.00$ | $\mathbf{0.93} \pm 0.00$ |
| | PSNR (↑) | $25.46 \pm 0.13$ | $\mathbf{26.39} \pm 0.16$ |
| Atari Boxing | LPIPS (↓) | $0.04 \pm 0.00$ | $\mathbf{0.02} \pm 0.00$ |
| | FVD (↓) | $237.90 \pm 12.15$ | $\mathbf{218.68} \pm 15.68$ |
| | SSIM (↑) | $0.93 \pm 0.00$ | $\mathbf{0.95} \pm 0.00$ |
| | PSNR (↑) | $\mathbf{30.00} \pm 0.14$ | $29.46 \pm 0.09$ |

## B.3 Evaluating World Model Accuracy

The world model accuracy for other procgen environments are shown in Table. 7 and Fig. 8 - Fig. 10. In most environments, Dyn-O significantly outperforms baselines in term of prediction accuracy.

Meanwhile, we further compare Dyn-O against DreamerV3 on the CLEVR dataset [24] and two ALE environments [4] (Skiing and Boxing), and the results are shown in Table. 8.

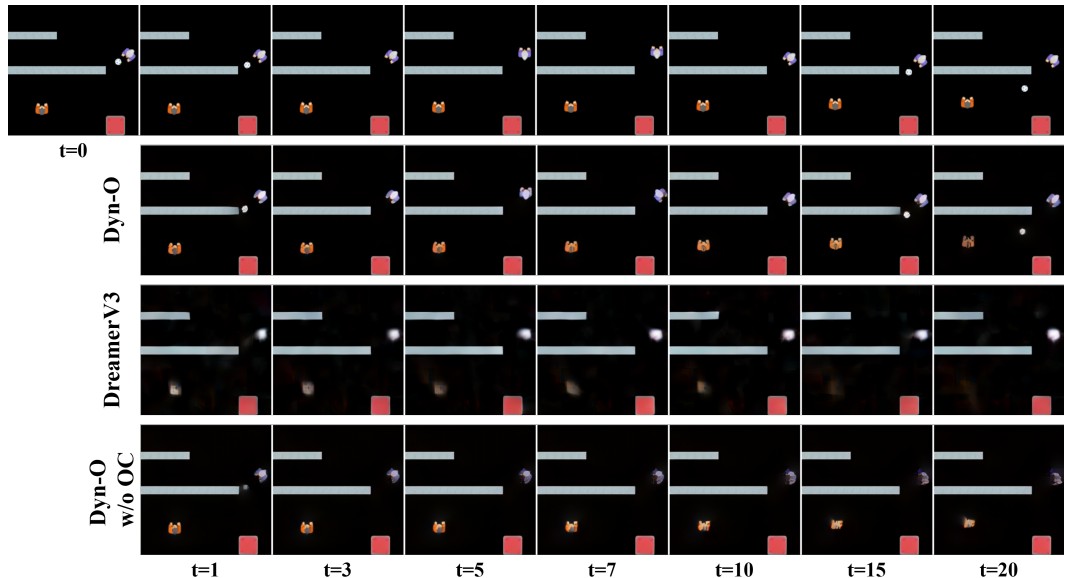

Figure 8: 20-step rollouts in **dodgeball**. **1st** row: ground-truth, **2nd** row: Dyn-O (ours), **3rd** row: DreamerV3, and **4th** row: Dyn-O w/o OC. Dyn-O significantly outperforms dreamer, with sharp player shape and accurate predictions of threw balls.

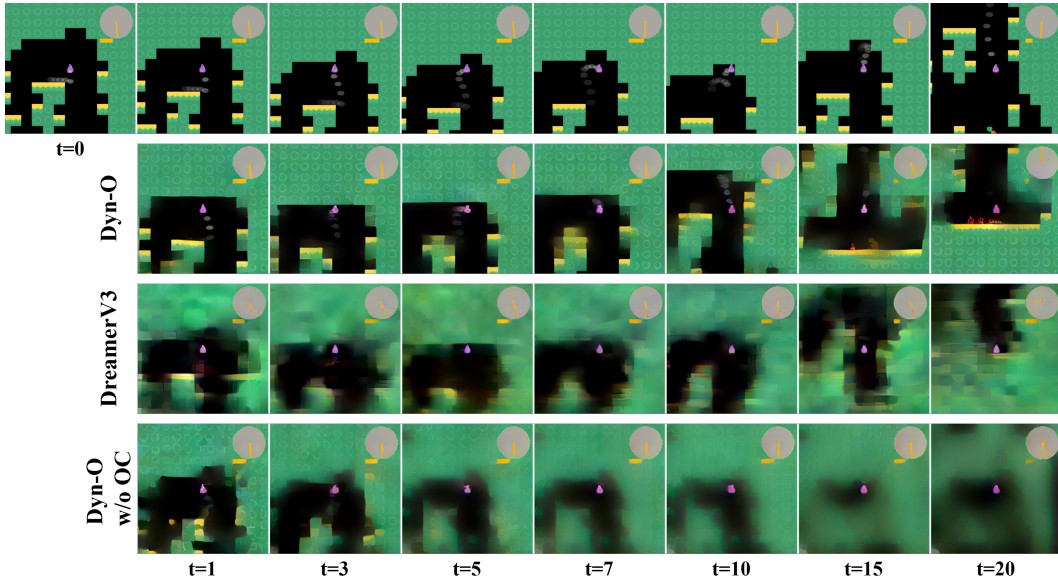

Figure 9: 20-step rollouts in **jumper**. **1st** row: ground-truth, **2nd** row: Dyn-O (ours), **3rd** row: DreamerV3, and **4th** row: Dyn-O w/o OC. Dyn-O significantly outperforms dreamer, with sharp wall and player trail until 10th timestamp.

## B.4 Evaluating Static-Dynamic Disentanglement

The probing accuracy for other procgen environments are shown in Table. 9 - 12. The same privilege information may belong to different properties in different environments, which we label out in the tables. In some environments, the same privilege information can be static properties of some objects and can dynamic for other objects. In such case, we mark such privilege information as "mixed".

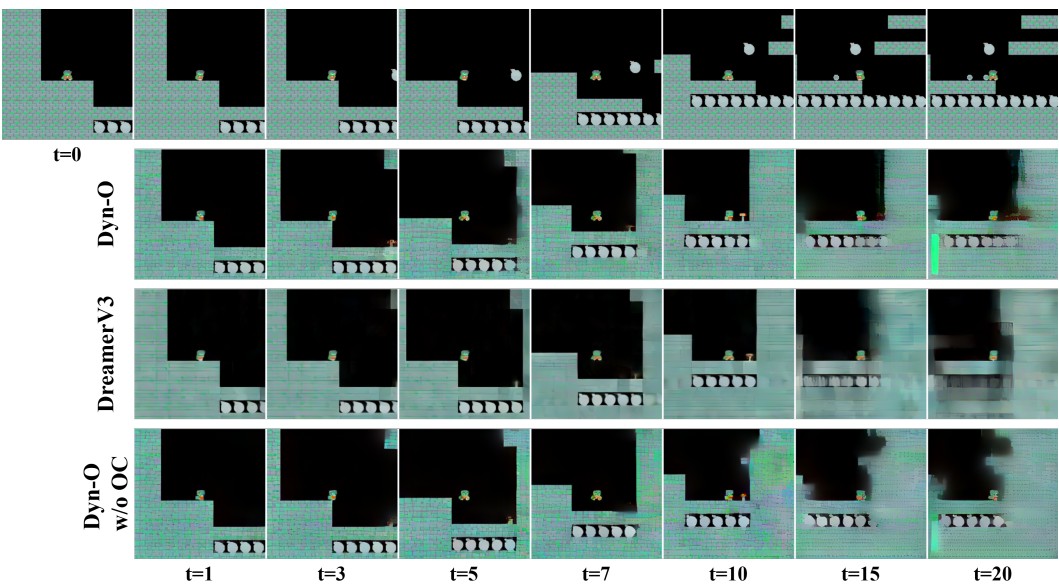

Figure 10: 20-step rollouts in **ninja**. **1st** row: ground-truth, **2nd** row: Dyn-O (ours), **3rd** row: DreamerV3, and **4th** row: Dyn-O w/o OC. Dyn-O significantly outperforms dreamer, with sharper wall shape at 15-th timestamp.

Table 9: Probing accuracy (↑), in percentage (%), on **bigfish** privilege properties.

|  | mean R values static | mean G values static | mean B values static | x position dynamic | y position static | area static |
|---|---|---|---|---|---|---|
| slots | 74.7 ± 0.0 | 77.9 ± 0.0 | 83.8 ± 0.0 | 97.7 ± 0.0 | 98.3 ± 0.0 | 100.0 ± 0.0 |
| dynamic features | 49.3 ± 3.0 | 48.4 ± 2.4 | 47.4 ± 3.7 | 94.0 ± 1.9 | 45.2 ± 5.1 | 95.9 ± 1.2 |
| static features | 65.8 ± 4.2 | 67.6 ± 4.8 | 77.5 ± 1.2 | 29.3 ± 0.4 | 88.8 ± 0.7 | 100.0 ± 0.0 |
| random features | 37.6 ± 0.0 | 31.8 ± 0.0 | 37.3 ± 0.0 | 19.2 ± 0.0 | 20.8 ± 0.0 | 88.6 ± 0.0 |


Table 10: Probing accuracy (↑), in percentage (%), on **dodgeball** privilege properties.

|  | mean R values static | mean G values static | mean B values static | x position mixed | y position mixed | area static |
|---|---|---|---|---|---|---|
| slots | 90.2 ± 0.0 | 91.7 ± 0.0 | 91.2 ± 0.0 | 98.7 ± 0.0 | 98.7 ± 0.0 | 99.8 ± 0.0 |
| dynamic features | 66.0 ± 1.6 | 62.4 ± 1.2 | 61.1 ± 1.8 | 55.4 ± 1.8 | 57.7 ± 2.5 | 95.3 ± 1.7 |
| static features | 73.1 ± 1.2 | 74.2 ± 1.4 | 75.5 ± 1.7 | 74.9 ± 2.4 | 70.4 ± 2.9 | 99.3 ± 0.0 |
| random features | 53.3 ± 0.0 | 35.5 ± 0.0 | 42.9 ± 0.0 | 20.1 ± 0.0 | 19.7 ± 0.0 | 90.1 ± 0.0 |

Table 11: Probing accuracy (↑), in percentage (%), on **ninja** privilege properties.

|  | mean R values static | mean G values static | mean B values static | x position dynamic | y position dynamic | area dynamic |
|---|---|---|---|---|---|---|
| slots | $87.1 \pm 0.0$ | $80.7 \pm 0.0$ | $86.1 \pm 0.0$ | $95.0 \pm 0.0$ | $96.6 \pm 0.0$ | $97.3 \pm 0.0$ |
| dynamic features | $60.7 \pm 9.7$ | $51.3 \pm 8.2$ | $60.6 \pm 9.0$ | $86.3 \pm 3.0$ | $88.1 \pm 0.3$ | $87.6 \pm 2.2$ |
| static features | $77.6 \pm 2.9$ | $62.2 \pm 0.4$ | $79.6 \pm 0.4$ | $35.9 \pm 0.9$ | $37.8 \pm 1.1$ | $82.2 \pm 1.3$ |
| random features | $32.9 \pm 0.0$ | $25.4 \pm 0.0$ | $31.3 \pm 0.0$ | $25.0 \pm 0.0$ | $19.5 \pm 0.0$ | $80.5 \pm 0.0$ |

Table 12: Probing accuracy (↑), in percentage (%), on **starpilot** privilege properties.

|  | mean R values static | mean G values static | mean B values static | x position dynamic | y position static | area static |
|---|---|---|---|---|---|---|
| slots | $71.5 \pm 0.0$ | $79.1 \pm 0.0$ | $71.0 \pm 0.0$ | $97.2 \pm 0.0$ | $97.5 \pm 0.0$ | $99.5 \pm 0.0$ |
| dynamic features | $55.6 \pm 7.2$ | $66.5 \pm 6.4$ | $55.3 \pm 6.2$ | $94.8 \pm 0.9$ | $77.0 \pm 14.2$ | $97.9 \pm 1.7$ |
| static features | $55.9 \pm 1.1$ | $70.7 \pm 1.1$ | $50.5 \pm 1.8$ | $26.8 \pm 0.7$ | $76.6 \pm 3.2$ | $99.2 \pm 0.0$ |
| random features | $35.9 \pm 0.0$ | $50.6 \pm 0.0$ | $36.3 \pm 0.0$ | $18.3 \pm 0.0$ | $22.7 \pm 0.0$ | $92.0 \pm 0.0$ |

- Please provide a short (1–2 sentence) justification right after your answer (even for NA).

**The checklist answers are an integral part of your paper submission.** They are visible to the reviewers, area chairs, senior area chairs, and ethics reviewers. You will be asked to also include it (after eventual revisions) with the final version of your paper, and its final version will be published with the paper.

The reviewers of your paper will be asked to use the checklist as one of the factors in their evaluation. While "[Yes] " is generally preferable to "[No] ", it is perfectly acceptable to answer "[No] " provided a proper justification is given (e.g., "error bars are not reported because it would be too computationally expensive" or "we were unable to find the license for the dataset we used"). In general, answering "[No] " or "[NA] " is not grounds for rejection. While the questions are phrased in a binary way, we acknowledge that the true answer is often more nuanced, so please just use your best judgment and write a justification to elaborate. All supporting evidence can appear either in the main paper or the supplemental material, provided in appendix. If you answer [Yes]  to a question, in the justification please point to the section(s) where related material for the question can be found.

IMPORTANT, please:

- **Delete this instruction block, but keep the section heading "NeurIPS Paper Checklist",**
- **Keep the checklist subsection headings, questions/answers and guidelines below.**
- **Do not modify the questions and only use the provided macros for your answers**.

