# OpenReview forum: "Dyn-O: Building Structured World Models with  Object-Centric Representations"
_NeurIPS.cc/2025/Conference — NeurIPS 2025 poster_

### Official Review · Reviewer_caxu · 2025-06-30

**Clarity:** 3
**Significance:** 2
**Originality:** 2
**Rating:** 4
**Confidence:** 4

**Summary:**

This paper proposes an object-centric world model for better capturing the dynamics of the environment. The world model is also designed to be able to decouple object features into dynamics-agnostic and dynamics-aware components. The authors conduct experiments on the challenging Procgen benchmark and demonstrate their method's superior prediction accuracy compared to DreamerV3.

**Questions:**

1.  The paper would be significantly more convincing if it included experiments evaluating the agent’s performance, either in a model-based or model-free setting. Currently, the focus on reconstruction accuracy leaves open whether the learned representations actually improve downstream policy learning. I would be willing to accept if this point is covered.

2. The paper evaluates the method on 7 Procgen environments but does not justify how these were chosen from the full benchmark suite. A brief explanation (e.g., diversity of difficulty, visual complexity, or relevance to object-centric learning) would clarify the experimental design.

**Ethical Concerns:**

["NO or VERY MINOR ethics concerns only"]

**Final Justification:**

See the comment

**Limitations:**

Yes.

**Paper Formatting Concerns:**

No major formatting issues.

**Quality:**

2

**Strengths And Weaknesses:**

**Strengths:**

1.  The paper incorporates pretrained computer vision models like SAM2 into world model learning. This is a promising direction, as leveraging large-scale pretrained models for representation learning in reinforcement learning remains underexplored. This work narrows the gap in this area.

2. The paper is well-structured, with clear illustrations and plots that effectively communicate the proposed method. The visualisations of reconstructed observations are helpful in understanding the model’s behaviour.

**Weakness:**

1. The primary goal of world models is to improve agent performance, yet the paper does not evaluate whether the learned representations actually benefit policy learning. While the method demonstrates improved reconstruction accuracy, this does not necessarily translate to better RL performance.
    - Is the static-dynamics disentanglement truly beneficial for agents?
    - If the agent can learn a good policy without such disentanglement, the additional complexity may be unjustified.
    - Could the gains simply come from increased model capacity?

    Without policy learning experiments, the practical utility of the proposed method remains unclear.



2. The experiments rely on trajectories generated by a pretrained PPG agent, raising concerns about applicability in online settings:
    - Does the method work in an online RL setting like DreamerV3? Online world models must learn from dynamically changing data distributions, and it is unclear whether the proposed approach would remain stable.
    - Could training collapse when learning from scratch? Pretrained policies provide high-quality exploration data, but in online RL, poor initial exploration might lead to degenerate representations.

3. The paper does not explicitly describe whether reconstruction metrics are computed on training or unseen test trajectories. Is the reported accuracy based on training rollouts? If so, the improvements could stem from overfitting rather than better generalisation.

---

> ### Author Rebuttal · Authors · 2025-07-31
>
> Thank you for your thoughtful and constructive comments. We’re glad that you recognize the potential of leveraging large-scale pretrained computer vision models like SAM2 within world model learning, and appreciate our efforts to narrow the gap in this underexplored area. We also thank you for acknowledging our paper’s well-organized presentation as well as its effective illustrations and visualizations. Below, we address your comments and questions.
>
> > `W1`: The experiments rely on trajectories generated by a pretrained PPG agent, raising concerns about applicability in online settings:
> > * Does the method work in an online RL setting like DreamerV3? Online world models must learn from dynamically changing data distributions, and it is unclear whether the proposed approach would remain stable.
> > * Could training collapse when learning from scratch? Pretrained policies provide high-quality exploration data, but in online RL, poor initial exploration might lead to degenerate representations.
>
> `R1`:
> * We would like to clarify that our focus is not on online reinforcement learning, but rather on building world models based on object-centric representations. In earlier experiments, we found that training the entire pipeline online was prohibitively resource-intensive, so we decided not to pursue that direction. Notably, many existing works also focus on the **offline setting** \[1-4\], so we believe our approach still offers meaningful contributions to the field.
>   We agree that investigating the stability of the proposed method under dynamically changing data distributions would be insightful. However, doing so would require substantial engineering efforts to significantly optimize our pipeline. We view this as a valuable direction for future work.
> * Thank you for the interesting question\! First, we would like to clarify that the training data we used were **not collected by an expert policy**, but rather by **different intermediate checkpoints** saved during RL training. As a result, the dataset contains a significant portion of suboptimal exploration data.
>   To further investigate this, we conducted an ablation experiment in the Bigfish environment, using only data collected from a **random policy** to train the representation. This was done to assess whether representation collapse would occur. The results are summarized in the table below. We can see using random exploration data leads to **slightly worse**, but not significantly degraded, performance.
>
>
> **Table 1**: Rollout accuracy of **Dyn-O trained with different data** at 20-th timestep in **Procgen Bigfish**, measured as mean and standard error
>
> | Data Collected by | LPIPS (↓) | FVD (↓) | SSIM (↑) | PSNR (↑) |
> | :---- | :---: | :---: | :---: | :---: |
> | PPG Policy Mixture | 0.25 ± 0.01 | 126.88 ± 6.42 | **0.76 ± 0.01** | 20.28 ± 0.30 |
> | Random Policy | 0.26 ± 0.02 | 137.90 ± 12.15 | 0.70 ± 0.01 | 20.17 ± 0.07 |
>
> We believe the extent to which representation quality is affected depends heavily on the properties of the environment. In many of the environments we use, most objects remain visible even under random policies, making representation collapse unlikely. However, in particularly challenging environments that require extensive exploration (e.g., Montezuma’s Revenge), such collapse may occur. That said, effective exploration in such environments is itself a broader and open research question, which we consider beyond the scope of our current work.
>
> \[1\] Jiang, Jindong, et al. "Object-centric slot diffusion." arXiv preprint arXiv:2303.10834 (2023).
> \[2\] Wu, Ziyi, et al. "Slotdiffusion: Object-centric generative modeling with diffusion models." Advances in Neural Information Processing Systems 36 (2023): 50932-50958.
> \[3\] Wu, Ziyi, et al. "Slotformer: Unsupervised visual dynamics simulation with object-centric models." arXiv preprint arXiv:2210.05861 (2022).
> \[4\] Singh, Gautam, Yi-Fu Wu, and Sungjin Ahn. "Simple unsupervised object-centric learning for complex and naturalistic videos." Advances in Neural Information Processing Systems 35 (2022): 18181-18196.
>
> > `W2`: The paper does not explicitly describe whether reconstruction metrics are computed on training or unseen test trajectories. Is the reported accuracy based on training rollouts? If so, the improvements could stem from overfitting rather than better generalisation.
>
> `R2`: We would like to point out that, **Line 236** of our paper describes that the evaluation is performed on unseen test levels. Therefore, the observed improvements are unlikely to result from overfitting.
>
> > `Q1`: The paper would be significantly more convincing if it included experiments evaluating the agent’s performance, either in a model-based or model-free setting. Currently, the focus on reconstruction accuracy leaves open whether the learned representations actually improve downstream policy learning. I would be willing to accept if this point is covered.
>
> `R3`: Thank you for the great suggestion and for your willingness to accept our paper if this point is addressed\! This is indeed an important aspect that could further strengthen our paper. During the rebuttal period, we conducted additional policy learning experiments as recommended. Specifically, to evaluate whether the learned representations facilitate policy learning, we **fixed the representation backbone and trained only the policy head using the PPG algorithm** on three Procgen games: **Bigfish**, **Starpilot**, and **Coinrun**. We used the “easy” difficulty setting with an unlimited number of levels.
>
> We compared the performance of **object-centric representations** against the representations directly output by the **Cosmos encoder**. Due to NeurIPS policy, we are unable to include figures at this stage, so we provide a table below summarizing the results at different training steps. Each game is trained with three seeds, and some runs are still in progress. We will update the results during the discussion phase once training is complete, and include the full figures in the final version. Based on the results so far, the learned representations lead to clearly improved policy learning performance.
>
> **Table 1**: Episodic returns in **Starpilot**
>
> | Steps | 2M | 4M | 6M | 10M | 25M |
> | ----- | :---- | :---- | :---- | :---- | :---- |
> | Object-centric | **7.81** | **15.06** | **17.13** | running | running |
> | Cosmos | 3.34 | 4.48 | 5.33 | 7.54 | running |
>
> **Table 1**: Episodic returns in **Bigfish**
>
> | Steps | 5M | 8M | 10M | 15M | 25M |
> | ----- | :---- | :---- | :---- | :---- | :---- |
> | Object-centric | **3.77** | **4.94** | running | running | running |
> | Cosmos | 0.95 | 0.84 | 0.88 | 0.91 | running |
>
> **Table 1**: Episodic returns in **Coinrun**
>
> | Steps | 5M | 8M | 10M | 15M | 25M |
> | ----- | :---- | :---- | :---- | :---- | :---- |
> | Object-centric | **8.74** | **8.44** | running | running | running |
> | Cosmos | 7.88 | 7.45 | 7.54 | 8.02 | running |
>
> > `Q2`: The paper evaluates the method on 7 Procgen environments but does not justify how these were chosen from the full benchmark suite. A brief explanation (e.g., diversity of difficulty, visual complexity, or relevance to object-centric learning) would clarify the experimental design.
>
> `R4`: Thank you for pointing this out. We selected 7 Procgen environments based on their **relevance to object-centric learning** and **visual complexity**. Specifically, we looked for environments with **moving objects**, **dynamic layouts**, and **minimal sensitivity to color**. Based on these criteria, we excluded maze-like environments such as Maze, Heist, Chaser, and Miner, as well as Plunder, where the ship's color plays a critical role. Among the remaining games, we randomly selected 7 while ensuring diversity across game types, given resource constraints. We would be happy to evaluate additional games if the reviewer deems it necessary, and we will include the above rationale in the next version.

---

> > ### Author Response · Authors · 2025-08-03
> > **[Q1] Result updates for online RL with fixed representation**
> >
> > > `Q1`: The paper would be significantly more convincing if it included experiments evaluating the agent’s performance, either in a model-based or model-free setting. Currently, the focus on reconstruction accuracy leaves open whether the learned representations actually improve downstream policy learning. I would be willing to accept if this point is covered.
> >
> > `R3`: Thank you for your patience and your willingness to accept our paper if this point is addressed\! We would like to update the resulting comparing **online RL performance with different fixed the representation backbones**.
> >
> > Even though the policies using our Dyn-O representation are only trained for 10/15M steps, their returns are already much higher than the policies using features from Comos encoder at 25M steps, suggesting the advantage of learning RL policies on top of object-centric representations.
> > We will include the full results in the final version of the paper.
> >
> > Table 1: Episodic returns in Procgen **Starpilot**, **Bigfish**, and **Coinrun**, shown as the average acorss 3 random seeds
> >
> > | Starpilot |  |  |  |  |  |
> > | ----- | :---- | :---- | :---- | :---- | :---- |
> > | Steps | 2M | 4M | 6M | 10M | 25M |
> > | Object-centric | **7.81** | **15.06** | **17.13** | **18.49** | running |
> > | Cosmos | 3.34 | 4.48 | 5.33 | 7.54 | 9.82 |
> >
> > | Bigfish |  |  |  |  |  |
> > | ----- | :---- | :---- | :---- | :---- | :---- |
> > | Steps | 5M | 8M | 10M | 15M | 25M |
> > | Object-centric | **3.77** | **4.94** | **5.74** | **7.70** | running |
> > | Cosmos | 0.95 | 0.84 | 0.88 | 0.91 | 0.97 |
> >
> > | Coinrun |  |  |  |  |  |
> > | ----- | :---- | :---- | :---- | :---- | :---- |
> > | Steps | 5M | 8M | 10M | 15M | 25M |
> > | Object-centric | **8.74** | **8.44** | **8.85** | **9.04** | running |
> > | Cosmos | 7.88 | 7.45 | 7.54 | 8.02 | 8.30 |
> >
> > **If our answers have addressed your question and confusion, we'd be grateful if you could revise the score.** Thank you for the careful reading and constructive comments! If there are any additional questions or issues that remain unresolved, we would be more than happy to provide further clarifications.

---

> > ### Comment · Reviewer_caxu · 2025-08-05
> >
> > I appreciate the authors' efforts in the rebuttal.
> >
> > Re: R2: It makes sense for this setting then. I think there should be some explanation for the level conception in Procgen, since the paper focuses on testing the performance of reconstruction accuracy, and the level is not a commonly used term in this field.
> >
> > Re: R3: To me, this result is good, and I hope the full results can be placed in the main text or the appendix later. I will raise my score to 4. Not higher due to this result is not presented when submitted, and both the reconstruction and agent performance analysis are not performed on the full Procgen benchmark.

---

> > > ### Author Response · Authors · 2025-08-05
> > >
> > > Thanks a lot for your constructive suggestions and for considering our rebuttal! We sincerely appreciate your feedback and will incorporate your comments and the additional results into the next version of the paper.

---

### Official Review · Reviewer_eNWE · 2025-06-30

**Clarity:** 3
**Significance:** 2
**Originality:** 3
**Rating:** 4
**Confidence:** 3

**Summary:**

This paper addresses the challenge of learning a world model that decomposes observations into static parts and dynamic objects. The proposed method proceeds as follows: First, it encodes images using the COSMOS tokenizer [A], obtaining a fixed set of K latents representing individual objects via slot attention [B]. These latents are then used to generate reconstructed images. The overall network is trained to minimize reconstruction error, a standard approach in object-centric representation learning [B; C]. The method further improves upon this framework by leveraging a pre-trained SAM2 model [Ravi et al., 2024], utilizing its segmentation masks as attention masks for slot attention to ensure accurate assignment of individual objects to their corresponding slots.

With this learned object-centric representation, the method trains a world model to predict, given a set of slots and actions, the next-frame slots, a reward, and the probability of episode termination.

The method is evaluated on the Procgen dataset [Cobbe et al., 2020], which comprises procedurally generated 2D video game environments.

[A] Agarwal, Niket, et al. "Cosmos world foundation model platform for physical ai." arXiv preprint arXiv:2501.03575 (2025).

[B] Locatello, F. et al. “Object-Centric Learning with Slot Attention.” Advances in Neural Information Processing Systems (NeurIPS), 2020.

[C] Engelcke, Martin, et al. "Genesis: Generative scene inference and sampling with object-centric latent representations." arXiv preprint arXiv:1907.13052 (2019).

**Questions:**

* The first sentence in the abstract is: "World models aim to capture the dynamics of the environment". Is this a fair statement? Intuitively, the first thing world model must learn is the static aspect of the world. Not to say learning dynamics isn't important -- it absolutely is -- but this emphasis doesn't sound right to me.

* Abstract states: "the dynamics are highly centered on interactions among objects within the environment". Could the rebuttal clarify this? I am confused by this statement.

**Ethical Concerns:**

["NO or VERY MINOR ethics concerns only"]

**Final Justification:**

After the rebuttal, I no longer have doubts regarding the scope of experimental validation, and I believe that the paper passes the bar.

**Limitations:**

The paper does not discuss limitations.

**Paper Formatting Concerns:**

No concerns.

**Quality:**

2

**Strengths And Weaknesses:**

### Strengths

* I like the paper’s premise that decomposing the static world and individual objects (i.e., learning a disentangled representation) facilitates effective world model learning.
* The methodology’s representation is clearly presented and straightforward to follow.
* The approach’s use of an object-centric latent space for world modeling is innovative and compelling, offering a novel perspective on structured representation learning.

### Weaknesses

* ***Experimental Validation***:
The paper states: “However, the development of object-centric world models has largely been explored in environments with limited visual complexity (such as basic geometries). It remains underexplored whether such models can be effective in more challenging settings.” I fully agree with this assessment; however, the evaluation on the Procgen dataset [A], which comprises procedurally generated 2D video game environments, mirrors the limited complexity of prior datasets [D]. Therefore, this paper's experimental validation does not address whether the proposed methodology is effective in more challenging, real-world environments. More critically, the method is tested solely on Procgen, without comparing its performance against baselines on datasets commonly used in prior work [B, C]. This omission prevents firm conclusions about whether the proposed methodology advances the field relative to existing approaches. Additionally, the world model’s experimental evaluation is limited beyond the object-centric representation. Could the authors clarify why broader datasets, such as the Arcade Learning Environment (ALE) [E], were not included?


* **Unclear Relation to Prior Art**:
The paper presents two core contributions: object-centric representations and the world model. However, it lacks clarity on how these components compare, conceptually and empirically, to prior work. While the paper references DreamerV3 as a baseline, it reports results only on Procgen, whereas DreamerV3 demonstrates performance across multiple datasets, Procgen, ALE [E] and several others with greater diversity than 2D games. This limited comparison undermines the paper’s claim of advancing the state of the art.

Overall, this work shows promise, but it requires significantly broader experimental validation to warrant presentation at a venue like NeurIPS.


References:

[A] Cobbe, K. et al. “Leveraging Procedural Generation to Benchmark Reinforcement Learning.” International Conference on Machine Learning (ICML), 2020.

[B] Greff, K. et al. “Multi-Object Representation Learning with Iterative Variational Inference.” International Conference on Machine Learning (ICML), 2019.

[C] Kabra, R. et al. “SIMONe: View-Invariant, Temporally-Abstracted Object Representations via Unsupervised Video Decomposition.” Advances in Neural Information Processing Systems (NeurIPS), 2021.

[D] Girdhar, R. and D. Ramanan. “CATER: A Diagnostic Dataset for Compositional Actions and Temporal Reasoning.” arXiv:1910.04744, 2019.

[E] Bellemare, M. G. et al. “The Arcade Learning Environment: An Evaluation Platform for General Agents.” Journal of Artificial Intelligence Research, 2013.

---

> ### Author Rebuttal · Authors · 2025-07-31
>
> Thank you for your valuable and insightful feedback. We're glad to hear that you found our methodology and representation clearly presented and easy to follow. We're especially encouraged that you consider our use of an object-centric latent space for world modeling to be innovative and a meaningful contribution to structured representation learning. Below, we address your comments and questions.
>
> > `W1`: Experimental Validation: The paper states: “However, the development of object-centric world models has largely been explored in environments with limited visual complexity (such as basic geometries). It remains underexplored whether such models can be effective in more challenging settings.” I fully agree with this assessment; however, the evaluation on the Procgen dataset \[A\], which comprises procedurally generated 2D video game environments, mirrors the limited complexity of prior datasets \[D\]. Therefore, this paper's experimental validation does not address whether the proposed methodology is effective in more challenging, real-world environments.
>
> `R1`: We agree with the reviewer that real-world videos are the best benchmark for evaluating object-centric world models. However, we would like to clarify that, while Procgen consists of 2D video game environments, it offers significantly greater visual and dynamic complexity compared to prior benchmarks such as **CATER** \[D\] and **CLEVR** \[2\]. Specifically, Procgen features:
>
> * **diverse shapes and textures** \- while most objects in prior benchmarks use a single color and basic shape, as shown in Fig 7 and 9, Procgen objects have more complex shapes (like spaceships, trails) and color combinations (like explosion).
> * **cluttered scenarios** \- in prior benchmarks, most scenarios have \<10 objects, while in Procgen, environments like Starpilot can have \~50 objects.
> * **complex dynamics** \- in prior benchmarks, most movements are constrained to linear motion with constant velocity. In contrast, Procgen features complex motions (like jump) and mechanisms (like starship explosion).
>
> Therefore, evaluation on Procgen meaningfully advances the study of object-centric world models toward more challenging settings, and exploring real-world videos remains an exciting avenue for future work.
>
> \[1\] Girdhar, R. and D. Ramanan. “CATER: A Diagnostic Dataset for Compositional Actions and Temporal Reasoning.” arXiv:1910.04744, 2019\.
> \[2\] Johnson, Justin, et al. "Clevr: A diagnostic dataset for compositional language and elementary visual reasoning." Proceedings of the IEEE conference on computer vision and pattern recognition. 2017\.
>
> > `W2`: More critically, the method is tested solely on Procgen, without comparing its performance against baselines on datasets commonly used in prior work \[B, C\]. This omission prevents firm conclusions about whether the proposed methodology advances the field relative to existing approaches. Additionally, the world model’s experimental evaluation is limited beyond the object-centric representation. Could the authors clarify why broader datasets, such as the Arcade Learning Environment (ALE) \[E\], were not included?
>
> `R2`: Thank you for the helpful feedback\! During the rebuttal period, we conducted additional experiments by running our Dyn-O model on the **CLEVR** dataset. Regarding ALE, we believe Procgen serves as an enhanced benchmark, offering comparable or even greater complexity, with the added advantage of being designed with **generalization** in mind. As a result, models trained on Procgen are **less prone to overfitting**. That said, we were able to run experiments on two ALE games, **Boxing** and **Skiing**, during the rebuttal period. Detailed results for both the CLEVR and ALE experiments are provided in our response to the next comment.
>
> > `W3`: Unclear Relation to Prior Art: The paper presents two core contributions: object-centric representations and the world model. However, it lacks clarity on how these components compare, conceptually and empirically, to prior work. While the paper references DreamerV3 as a baseline, it reports results only on Procgen, whereas DreamerV3 demonstrates performance across multiple datasets, Procgen, ALE \[E\] and several others with greater diversity than 2D games. This limited comparison undermines the paper’s claim of advancing the state of the art.
>
> `R3`: Thank you for the suggestion\! Following your recommendation, we conducted experiments comparing Dyn-O and DreamerV3 on **CLEVR** and two **ALE** games. As the results below show, Dyn-O outperforms DreamerV3 in ALE, highlighting the advantages of object-centric world models. The experiments on CLEVR are currently in progress, and we will update the results as soon as they are available.
>
> **Table 1**: Rollout accuracy at 20-th timestamp in **CLEVR**, measured as mean and standard error
>
> | Method | LPIPS (↓) | FVD (↓) | SSIM (↑) | PSNR (↑) |
> | :---- | :---: | :---: | :---: | :---: |
> | Dyn-O | running |  |  |  |
> | DreamerV3 | running |  |  |  |
>
> **Table 2**: Rollout accuracy at 20-th timestamp in **Atari Skiing**, measured as mean and standard error
>
> | Method | LPIPS (↓) | FVD (↓) | SSIM (↑) | PSNR (↑) |
> | :---- | :---: | :---: | :---: | :---: |
> | Dyn-O | **0.09 ± 0.01** | **187.76 ± 8.04** | **0.93 ± 0.00** | **26.39 ± 0.16** |
> | DreamerV3 | 0.12 ± 0.00 | 217.84 ± 3.76 | 0.91 ± 0.00 |  25.46 ± 0.13 |
>
> **Table 3**: Rollout accuracy at 20-th timestamp in **Atari Boxing**, measured as mean and standard error
>
> | Method | LPIPS (↓) | FVD (↓) | SSIM (↑) | PSNR (↑) |
> | :---- | :---: | :---: | :---: | :---: |
> | Dyn-O | **0.02 ± 0.00** | **218.68 ± 15.68** | **0.95 ± 0.00** | 29.46 ± 0.09 |
> | DreamerV3 | 0.04 ± 0.00 | 237.90 ± 12.15 | 0.93 ± 0.00 | **30.00 ± 0.14** |
>
> > `Q1`: The first sentence in the abstract is: "World models aim to capture the dynamics of the environment". Is this a fair statement? Intuitively, the first thing world model must learn is the static aspect of the world. Not to say learning dynamics isn't important \-- it absolutely is \-- but this emphasis doesn't sound right to me.
>
> `R4`: We agree that a world model can capture both the dynamic aspects (transition dynamics) and the static aspects (representation) of an environment. We wrote this statement because most prior work on world models has focused on the dynamic aspect, while the representation side, such as learning object segmentations and image understanding, has been more extensively explored in the broader deep learning and computer vision communities. We are happy to revise this statement based on your suggestions.
>
> > `Q2`: Abstract states: "the dynamics are highly centered on interactions among objects within the environment". Could the rebuttal clarify this? I am confused by this statement.
>
> `R5`: Thank you for your feedback\! By this statement, we mean that most changes in the environment (its dynamics aspect), are either caused by objects or affect them. For example, in the Procgen Coinrun game, changes in the game screen are primarily driven by the agent’s movement or interactions with other objects, such as collecting coins, getting hurt by enemies, or being blocked by chests. We are happy to revise the statement according to your suggestions.

---

> > ### Author Response · Authors · 2025-08-03
> > **[W3] CLEVR result update**
> >
> > > `W2`: More critically, the method is tested solely on Procgen, without comparing its performance against baselines on datasets commonly used in prior work. This omission prevents firm conclusions about whether the proposed methodology advances the field relative to existing approaches.
> >
> > `R2`: Thank you for the suggestion and patience! In addtion to the ALE results shown above, we would like to update the results in **CLEVR**, one of datasets commonly used in prior work. As shown below, Dyn-O significantly outperforms Dreamweaver in terms of rollout accuracy, suggesting the advantage of learning dynamics models on top of object-centric representations.
> >
> > **Table 1**: Rollout accuracy at 20-th timestamp in **CLEVR**, measured as mean and standard error
> >
> > | Method | LPIPS (↓) | FVD (↓) | SSIM (↑) | PSNR (↑) |
> > | :---- | :---: | :---: | :---: | :---: |
> > | Dyn-O | **0.31 ± 0.00** | **446.57 ± 8.55** | **0.88 ± 0.00** | **22.63 ± 0.06** |
> > | DreamerV3 | 0.34 ± 0.00 | 1676.18 ± 40.92 | 0.86 ± 0.00 | 21.37 ± 0.09 |
> >
> > **If our answers have addressed your question and confusion, we'd be grateful if you could revise the score.** Thank you for the careful reading and constructive comments! If there are any additional questions or issues that remain unresolved, we would be more than happy to provide further clarifications.

---

> > > ### Comment · Reviewer_eNWE · 2025-08-05
> > >
> > > Dear authors,
> > >
> > > I appreciate clarifications and expanded results. I do have one more concern, it appears that the proposed method is using several components, trained on large amounts of data (e.g., SAM), compared to methods trained on target datasets only.
> > >
> > > Could you provide a summary of the components / trained on which the proposed method and baselines are trained? Ideally, such a table should be in the paper to contextualize these differences. I would like to have an overview of this before making a final decision.

---

> > > > ### Author Response · Authors · 2025-08-05
> > > > **Components of each method and their training data**
> > > >
> > > > Thanks for considering our rebuttal and providing constructive feedback\!
> > > >
> > > > We would like to clarify that, as mentioned in **Line 213**, for each procgen environment, all methods are trained on the **same** dataset. The only difference is that, during object-centric representation learning, Dyn-O additionally uses SAM to extract the segmentation mask for each image in the dataset, which warms up representation learning and improves representation quality. You can find the details of the components and datasets below.
> > > >
> > > > **Table 4**: World model results \- components of each method and their training data
> > > >
> > > > | Method | Component | Training Data |
> > > > | :---- | :---- | :---- |
> > > > | Dyn-O | **fixed** Cosmos encoder-decoder | N/A |
> > > > |  | slot attention (for object-centric features) | images \+ segmentation masks from SAM |
> > > > |  | world model (backbone: state-space models) | transitions |
> > > > | Dyn-O w/o OC | **fixed** Cosmos encoder-decoder | N/A |
> > > > |  | world model (backbone: state-space models) | transitions |
> > > > | DreamerV3 | **fixed** Cosmos encoder-decoder | N/A |
> > > > |  | world model (backbone: RNN) | transitions |
> > > >
> > > > **Table 5**: Object-centric representation results \- components of each method and their training data (the world model is omitted as it’s not used for representation evaluation)
> > > >
> > > > | Method | Component | Training Data |
> > > > | :---- | :---- | :---- |
> > > > | Dyn-O | **fixed** Cosmos encoder-decoder | N/A |
> > > > |  | slot attention | images \+ segmentation masks from SAM |
> > > > | SOLV | **fixed** Cosmos encoder-decoder | N/A |
> > > > |  | slot attention | images |
> > > >
> > > > If there are any additional questions or issues that remain unresolved, we would be more than happy to provide further clarifications.

---

### Official Review · Reviewer_EH1k · 2025-07-02

**Clarity:** 3
**Significance:** 3
**Originality:** 3
**Rating:** 4
**Confidence:** 3

**Summary:**

The paper introduces a new method for learning world model with object-centric representations. Given an input video, the method first detect objects in the video, and then extracts latent feature representations are each object. These latent representations are used in a state-space model to predict their next representations given an action. The method further decomposes the object latent representations into a static part and a dynamic part. Experiments are conducted on the Procgen dataset to verify the world model.

**Questions:**

Please see weaknesses above regarding questions.

**Ethical Concerns:**

["NO or VERY MINOR ethics concerns only"]

**Final Justification:**

The authors provided more experimental comparison and ablation studies in their rebuttal.

**Limitations:**

No limitations discussed in the paper.

**Quality:**

3

**Strengths And Weaknesses:**

Positives:

-	The paper has introduced several novel ideas in my opinion such as using the state-space model for prediction, and decomposition of a static and a dynamic components of the object slot representations.

-	The paper also carefully designs what loss functions to use for learning the world model.

-	The experimental results show improvement compared to DreamerV3.

Negatives:

-	The experimental evaluation of the method seems not very thorough. For example, the comparison is only against DreamerV3. There is no other world model evaluated.

-	The experimental environments are all about games. More sophisticated setups can be considered such as using real-world videos.

-	Some ablation studies can be added. For example, how important is using the state-space model? What if some other models are used?

---

> ### Author Rebuttal · Authors · 2025-07-31
>
> We sincerely appreciate your insightful comments. We're glad to hear that you find our ideas novel and recognize the careful design of the loss functions. Below, we address the questions you raised.
>
> > `W1`: The experimental evaluation of the method seems not very thorough. For example, the comparison is only against DreamerV3. There is no other world model evaluated
>
> `R1`: Thank you for the suggestion\! During the rebuttal period, we further compared our Dyn-O with **Dreamweaver** \[1\], a recent object-centric world model. The experiments are currently in progress, and we will update the results as soon as they are available.
>
> \[1\] Baek, Junyeob, et al. "Dreamweaver: Learning Compositional World Models from Pixels." arXiv preprint arXiv:2501.14174 (2025).
>
> > `W2`: The experimental environments are all about games. More sophisticated setups can be considered such as using real-world videos.
>
> `R2`: We agree that evaluating on real-world videos would further demonstrate the applicability of our object-centric world model and represents an exciting direction for future work. One avenue we plan to explore is evaluating Dyn-O on **robotic datasets** such as Bridge-v2 \[2\] and RT-1 \[3\] to assess its performance in more realistic settings. However, due to time and compute constraints, we were unable to include these experiments during the rebuttal period. We intend to pursue them in future extensions of this work.
>
> \[2\] Walke, Homer Rich, et al. "Bridgedata v2: A dataset for robot learning at scale." Conference on Robot Learning. PMLR, 2023.
> \[3\] Brohan, Anthony, et al. "Rt-1: Robotics transformer for real-world control at scale." arXiv preprint arXiv:2212.06817 (2022).
>
> > `W3`: Some ablation studies can be added. For example, how important is using the state-space model? What if some other models are used?
>
> `R3`: Thank you for the suggestion\! We conducted an ablation study on the backbone model for Dyn-O in the **Bigfish** environment, replacing the **state-space model** (SSM) with a **transformer**. As shown below, the SSM significantly outperforms the transformer.
>
> **Table 1**: Rollout accuracy at 20-th timestep in **Bigfish**, measured as mean and standard error
>
> | Method | LPIPS (↓) | FVD (↓) | SSIM (↑) | PSNR (↑) |
> | :---- | :---: | :---: | :---: | :---: |
> | Dyn-O SSM | **0.25 ± 0.01** | **126.88 ± 6.42** | **0.76 ± 0.01** | **20.28 ± 0.30** |
> | Dyn-O Transformer | 0.39 ± 0.00 |  235.09 ± 3.23 | 0.73 ± 0.00 | 17.78 ± 0.08 |

---

> > ### Author Response · Authors · 2025-08-03
> > **[W1] Dreamweaver result update**
> >
> > > `W1`: The experimental evaluation of the method seems not very thorough. For example, the comparison is only against DreamerV3. There is no other world model evaluated.
> >
> > `R1`: Thanks for your patience! We would like to update the resulting comparing our Dyn-O against **Dreamweaver [1]**, a recent object-centric world model. As shown below, Dyn-O significantly outperforms Dreamweaver in terms of rollout accuracy.
> >
> > Rollout accuracy at 20-th timestamp in **Bigfish**, measured as mean and standard error
> >
> > | Method | LPIPS (↓) | FVD (↓) | SSIM (↑) | PSNR (↑) |
> > | :---- | :---: | :---: | :---: | :---: |
> > | Dyn-O | **0.25 ± 0.01** | **126.88 ± 6.42** | **0.76 ± 0.01** | **20.28 ± 0.30** |
> > | Dreamweaver | 0.52 ± 0.01 | 831.17 ± 15.55 | 0.68 ± 0.00 | 19.09 ± 0.12 |
> >
> > Rollout accuracy at 20-th timestamp in **Starpilot**, measured as mean and standard error
> >
> > | Method | LPIPS (↓) | FVD (↓) | SSIM (↑) | PSNR (↑) |
> > | :---- | :---: | :---: | :---: | :---: |
> > | Dyn-O | **0.22 ± 0.01** | **211.27 ± 12.70** | **0.76 ± 0.01** | **20.55 ± 0.13** |
> > | Dreamweaver | 0.50 ± 0.00 | 735.24 ± 21.48 | 0.68 ± 0.00 | 19.48 ± 0.13 |
> >
> > \[1\] Baek, Junyeob, et al. "Dreamweaver: Learning Compositional World Models from Pixels." arXiv preprint arXiv:2501.14174 (2025).

---

### Official Review · Reviewer_AdL1 · 2025-07-02

**Clarity:** 2
**Significance:** 3
**Originality:** 2
**Rating:** 4
**Confidence:** 4

**Summary:**

In this paper the authors present the Dyn-O model that learns world models by both learning object representations and a dynamics model to transition between those representations. Crucially, Dyn-O segments object representations into static properties that cannot change (e.g., the object sprite) and dynamics properties that can (e.g., position). Dyn-O is tested on seven different Procgen environments, outperforming DreamerV3 – an alternate world learning model – and on one environment disentangles the static and dynamic features appropriately

**Questions:**

My main question is how the findings differ and/or generalize across the different Procgen environments. For instance, Table 1 reports results aggregated for all environments, but it would be useful to assess how Dyn-O performs vs Dreamer or the non-object-centric version across all environments. Similarly, evaluating the object-centric representations is only done on BigFish and the static/dynamic disentanglements only on CoinRun – how do these findings generalize to the other environments?

**Ethical Concerns:**

["NO or VERY MINOR ethics concerns only"]

**Final Justification:**

In light of the author's responses to me and other reviewers that provided more results, I am raising my review to a 4. I still have some concerns about the limitations of the problem set tested (e.g., some of the results in the responses are on subsets of the games -- possibly due to the limited response period) but the extensions here have started to address them

**Limitations:**

Yes

**Quality:**

2

**Strengths And Weaknesses:**

This work presents an interesting idea for world model, especially in segmenting out the static and dynamic features to better capture real-world object structure. While relatively simple environments, using seven Procgen tasks is good for capturing diverse dynamics. However, the results report data that is either only aggregated across environments, or listed for a single environment at a time, making it difficult to assess how general the results are across different types of environments and dynamics

---

> ### Author Rebuttal · Authors · 2025-07-31
>
> Thank you for your valuable comments and feedback\! We're glad to hear that you found our idea interesting. Below, we address your questions.
>
> > `Q1:` My main question is how the findings differ and/or generalize across the different Procgen environments. For instance, Table 1 reports results aggregated for all environments, but it would be useful to assess how Dyn-O performs vs Dreamer or the non-object-centric version across all environments. Similarly, evaluating the object-centric representations is only done on BigFish and the static/dynamic disentanglements only on CoinRun – how do these findings generalize to the other environments?
>
> `R1:` Due to space limit, the results for each Procgen environment are deferred to the **supplementary materials**. Specifically,
>   * **Dynamics rollout accuracy**: As shown in **Table 6** (Page 15), Dyn-O outperforms both Dreamer and the non-object-centric baseline in most environments. Qualitative results are provided in Fig. 8 \- 10\.
>   * **Object-centric representation quality**: As shown in **Table 5** (Page 14), Dyn-O outperforms SOLV in 3 out of 4 environments. Corresponding qualitative results can be found in Fig. 7\.
>   * **Static/dynamic disentanglement**: As shown in **Tables 7 \- 10**, Dyn-O successfully disentangles static and dynamic features. In particular, for static privileged properties, static features achieve significantly higher probing accuracy than dynamic features, and vice versa.
>
> We will add clarification in the main text to help reduce potential confusion.

---

### Note · Authors · 2025-08-12

We apppreciate the AC and all reviewers' time and efforts to review our paper and providing constructive feedback!

In the final remarks, we would like to summarize how we have addressed each reviewer's questions:
* For R1 (AdL1), we clarify that the results for each Procgen environment are in the supplementary materials, where Dyn-O outperforms baselines in most environments.
* For R2 (EH1k),
  * We showed that **Dyn-O outperforms an addition baseline Dreamweaver**.
  * We evaluated an ablation of Dyn-O, where **state-space model backbones is significantly better than the transformer backbone**.
* For R3 (eNWE),
  * We showed that Dyn-O outperforms the baseline Dreamweaver on **datasets used by prior works (CLEVR and Atari)**.
  * We clarified that Dyn-O uses the **same training data** as the baselines.
* For R4 (caxu),
  * We showed that, compared to learnining from data collected by RL policies, when using data collected by a random policy, Dyn-O still performs comparably.
  * We showed the **object-centric representation learned by Dyn-O improves online RL sample efficiency and performance**, compared to learning from presentations by a pretrained Cosmos encoder.

If our answers have addressed your question and confusion, we'd be grateful if you could consider them.

---

### Decision · Program_Chairs · 2025-09-17

**Decision:**

Accept (poster)

**Comment:**

The paper introduces Dyn-O, a structured world model built on object-centric representations that disentangle static and dynamic features. Experimental results indicate that the approach improves rollout prediction and compositional generalization in visually complex environments.

Reviewers found the core idea innovative and the paper well-written, but significant concerns were raised due to limited experimental validation—confined mostly to the Procgen benchmark and a lack of evidence that the model's improved prediction accuracy translated to better performance on any downstream reinforcement learning task. The rebuttal was critical to alleviate these major concerns, and all reviewers were in agreement towards acceptance. The AC agrees with the consensus.

The authors are highly encouraged to incorporate the expanded evaluation (comparisons to new baselines and on different datasets, demonstration of improved sample efficiency, etc) in the camera ready version of the paper.